# Linear infrastructure drives biotic homogenization among bird species of a tropical dry forest

**Dishane K. Hewavithana**[1]*, **Devaka K. Weerakoon**[2], **Mayuri R. Wijesinghe**[2],
**Christopher A. Searcy**[1]*

**1** Abess Center for Ecosystem Science and Policy, University of Miami, Coral Gables, Florida, United States of America, **2** Department of Zoology and Environment Sciences, University of Colombo, Sri Lanka

* dkh48@miami.edu (DKH); cas383@miami.edu (CAS)

## Abstract

Linear infrastructures (LIs) such as roads, railroads, and powerlines are expanding rapidly around the globe. While most future developments are projected to take place in tropical regions, available information on impacts of LIs is biased towards single species studies of solely road impacts in temperate regions. Therefore, we investigated impacts of three types of LIs (road, railroad, and powerline) on the bird community of a tropical dry forest. Point-count surveys to record avian richness and abundance were conducted at 80 plots that were spatially stratified to include sites proximate to all possible LI combinations. Five measures of vegetation structure were collected at each plot as well. We then assessed the relationship between the bird community (i.e., richness, abundance, composition) and distance to each LI type while accounting for variation in vegetation structure. Species richness and abundance both declined significantly (26% and 20%, respectively) from edge habitat next to railroad to interior forest plots, while community composition was significantly altered by the distance to all three LIs. Road and railroad (both forms of dynamic infrastructure with moving vehicles) had similar effects on the bird community that contrasted with those of powerline (a type of static infrastructure). The resulting ordination reveled that Sri Lankan endemics are significantly disfavored by LI proximity, while species that now have naturalized populations across the globe are most often found proximal to LI. Our results emphasize that LI drive biotic homogenization by favoring these increasingly widespread species at the expense of unique elements of the biota.

## 1. Introduction

### 1. Types of linear infrastructure and their effects

Continuous expansion of the already extensive linear infrastructure (LI) network affects ecosystems around the world. More than 40% of the earth's total terrestrial surface, including numerous ecosystems and species, is located within 5 km of a road [1] and yet proposals for continued expansion proliferate. In the contiguous United States, more than 80% of forest land is already located within 1 km of a road [2]. Moreover, scientists predict a 60% global

**Data availability statement:** The data for this study can be found at https://doi.org/10.5061/dryad.kh18932hc.

**Funding:** Dr. Devaka K. Weerakoon and Dr. Dishane Hewavithana received funding from the National Research Council of Sri Lanka. Grant Number 16-029. The funders did not play any role in the study design, data collection and analysis, decision to publish, or preparation of the manuscript.

**Competing interests:** The authors have declared that no competing interests exist.

increase in road and rail networks by 2050 [3]. Utility easements and industrial corridors, such as powerlines, pipelines, and seismic exploration lines, are also expanding at a rapid rate [4]. All these different LI types are often used in tandem to provide access and energy, supporting a region's economic growth and development while traversing long distances through natural landscapes.

Due to their unique designs and functions, different LIs have the potential to differentially affect wildlife [5,6]. For instance, transport corridors, such as roads and railroads, exert a dynamic disturbance on the environment due to vehicular movement resulting in animal-vehicle collisions and emission of pollutants at the site. Even among LIs that exert dynamic impacts, effects may differ based on frequency and speed of vehicular movement [7,8]. Comparatively, the impact of LIs such as powerlines and pipelines are more static (no moving objects along the linear track). In this case, it is the presence of the infrastructure that affects wildlife by introducing artificial structures to the natural environment that could interfere with ecological interactions or animal behavior such as predation, perching, nesting, etc., [9,10].

Infrastructures that differ in their dynamic vs. static nature may still alter the physical environment in similar ways. For example, railroads and powerlines both create narrower corridors than multi-lane roads [5]. This can favor certain species, such as cottontail rabbits (*Sylvilagus transitionalis*), an early successional habitat specialist of conservation concern, that use such narrow corridors for movement among their habitat patches [11]. Additionally, all linear corridors cause habitat fragmentation and thus create edge effects [4,12]. For example, roads can affect fauna, flora, and the abiotic environment as far as 1 - 1.5 km into neighboring habitat [13,14]. However, we know little about the extent of edge effects of different types of LIs. Understanding the varying types and scales of impact from LI enables informed decision making, leading towards sustainable development.

## 2.  Gaps in LI research

There are several considerable gaps in information related to impacts of LI. These include biases with regard to the types of LI, the types of impacts, and the geographic regions that are typically studied. First, roads are the most extensively studied LI type around the world [4,5]. Roads cause direct mortality, habitat alteration, habitat fragmentation, and create barriers [15]. Although other LI types could potentially cause similar impacts to roads, only a limited number of studies have investigated these impacts [4,16]. Second, available literature on impacts of LI is skewed toward reporting direct mortality, while fewer studies report LI affecting biodiversity via other pathways such as changing animal behavior [17–19], spreading invasive species [20–22], and altering species interactions [23–25]. Third, the majority of scientific work conducted on LI has occurred in temperate ecosystems [4,26]. However, given the substantial differences in faunal and floral assemblages between tropical and temperate regions, studies from temperate systems may not accurately predict LI impacts in all circumstances. For instance, tropical regions harbour a great deal of endemic species, which are prone to replacement by generalist species, driving functional-homogenization [27]. A review of tropical forest studies revealed that forest fragmentation is the primary driver of such biotic homogenization (Freitag Kramer et al. 2023). LI is often the fragmenting agent, and biotic homogenization has been specifically documented in association with the expansion of high-speed railway networks (Falcão Rodrigues et al. 2024). Homogenized communities can be identified by a preponderance of habitat generalists compared to habitat specialists (Devictor et al. 2008).

Conservation and safety issues encountered due to LI also differ between tropical and temperate zones. For example, in temperate regions, colliding with large mammals is a greater

issue for both humans and animals [28,29], whereas in tropical regions roadkills mostly consist of amphibians and reptiles [30,31]. In addition, since many tropical regions are in developing nations, there is a pressing need for infrastructure development. The infrastructure expansion mentioned earlier expects to add 25 million kilometers of new paved roads globally by 2050, and 90% of this new infrastructure is going to be built in tropical regions [32]. Therefore, it is crucial to expand knowledge of the impacts of LI in tropical regions.

## 3. Birds as an indicator taxon

All four terrestrial vertebrate classes are directly affected by habitat loss and disturbance due to construction and operation of LI [14,16]. However, due to greater mobility (and therefore the ability to respond promptly to a given disturbance), as well as relatively high detectability, birds are ideal indicators to study the effects of LI [33]. For a given group of animals to be used as an indicator, they first should respond to the impact of interest. Birds are vocal animals and hence are likely to be disturbed by the noise created by both roads and railroads [34–36]. In addition, certain bird species prefer to travel under forest cover rather than cross open areas, even when the forested route is substantially longer than the short cut in the open [37]. Therefore, gaps created in their habitat are likely to affect their movement patterns, possibly acting as a barrier for certain species [38,39]. Second, the indicator group should be common enough at the study site to provide a robust sample size. At the site selected for the present study (Fig 1), birds are the most species-rich vertebrate group [40].

Therefore, to fill this crucial gap in information on impacts of different LI types in tropical regions, the present study attempts to quantify the spatial scale and relative magnitude of impacts from roads, railroads, and powerlines on the bird community in a tropical dry forest in

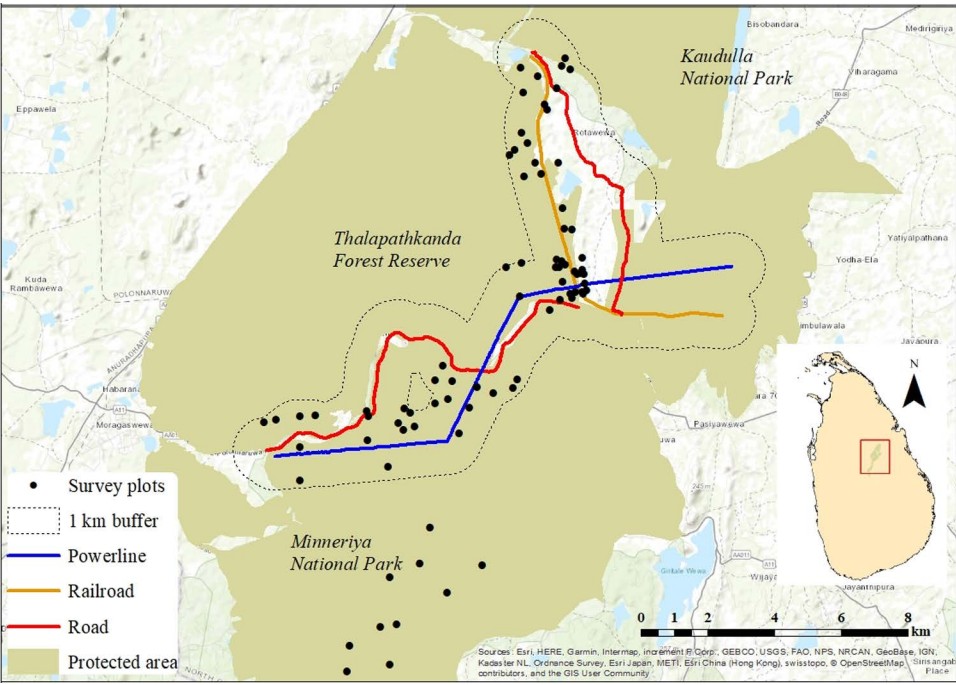

**Fig 1. Map of the study area showing the locations of the 80 survey points.** The buffer region from which the first 70 survey points were selected is outlined in black, with the road (red line), railroad (orange line), and powerline (blue line) depicted within. The ten remaining survey points are to the south located in interior forest.

Sri Lanka. The impact is quantified by looking at changes in bird species richness, abundance, and community composition in relation to distance from the selected LIs. In particular, we predict the following: 1) bird species richness and abundance will increase as distance to LIs increases, 2) bird community composition will vary with distance from LIs, with disturbance-tolerant species found closer to infrastructures and more sensitive species present farther away, and 3) type of LI will influence the impact on bird communities, with roads and railroads (dynamic impacts) having more similar effects compared to powerlines (static impacts).

## 2.  Materials and methods

### 1.  Study site

The study site was located in the Polonnaruwa District, North Central Province, Sri Lanka (8.07°N, 80.88°E). We selected this site for three main reasons. First, according to the National Physical Plan [41], this area will become the hub for four major urban centers by 2050. As a result, the area is scheduled for substantial expansions of its infrastructure network. Second, the forest ecosystems present in this region are part of the largest network of dry zone protected areas in the country. This provides an opportunity to investigate the current impact of LI networks on dry forest biodiversity so that these relationships can be considered during the planned infrastructure network expansion. Third, the selected site has all three LIs of interest (i.e., road, railroad, and powerline) in close proximity to each other within the same forested landscape.

The landscape consists of two nationally protected areas, Thalapathkanda Forest Reserve and Minneriya National Park. The main road traversing the site (AA011-Maradankadawela-Habarana- Thirukkondaiadimadu Road) connects two cities, Habarana and Polonnaruwa. The study road carries an average of ~ 10,800 vehicles per 24 hr, of which 28% are heavy vehicles (buses or trucks), 55% are passenger cars and vans, 14% are two-wheeled motor vehicles, and 3% are safari jeeps (Hewavithana et al., unpublished data). The railroad runs from Gal Oya to Polonnaruwa with a daily traffic volume of six trains (from station records). The powerline (132 kV) runs from Habarana to the Polonnaruwa 132/33 kV Grid substation. These LIs have all been in place for > 20 years. Hence, we assume that the fauna and flora at the site have had adequate time to respond to most of the infrastructure's effects.

The major habitat type found in this area is Tropical Dry-Mixed Evergreen Forest, the largest natural terrestrial habitat type in the country [42]. This habitat experiences a tropical dry hot monsoon climate characterized by a bimodal pattern of rainfall with relatively uniform high temperatures throughout the year (mean temperature of 28°C). The main rainy season occurs between October and January coinciding with the north-east monsoon. The area experiences desiccating winds between May and August during the long, hot south-west monsoon. The average altitude of the monitored locations is 187 m above sea level.

### 2.  Bird surveys

To determine the distribution of avian biodiversity across the landscape, we selected 80 survey points. The locations for these survey points were selected by marking the footprint of each LI through the forested landscape using ArcGIS 10.5.1 and then generating a 1 km buffer around all three LIs and merging them into a single region. We then selected 70 random points at least 100 m apart (mean distance to nearest neighboring survey point =  386 m) [43] within the resulting buffered region such that ten points fell within each of the seven distance categories outlined in Table 1. Ten additional survey points were selected within the study area >1 km from all of the LIs. We used the point-count method to determine the abundance of birds at each survey site [44], and species richness was estimated based on the incidence (presence/

absence) of species recorded at each plot [45]. All 80 survey points were surveyed three times over the course of the study: dry season (surveys conducted Jul-Aug 2019), wet season (surveys conducted Dec 2019-Jan 2020), and dry season (surveys conducted July-Aug 2020).

In this study, we used the bird species list from the National Red List of Sri Lanka as our source to determine species endemism [46]. Similarly, we used the Bird Life International database [47]to determine whether each species had naturalized populations outside of its native range. For species not included in this database, a Google Scholar search was conducted to find the same information. Warakagoda et al., [48] was consulted to categorize recorded species according to their relative forest dependency. The following keywords/phrases were used to determine if a given species prefers interior forest canopy: "roosts high in trees", "shy", "deep forest", "hidden in forest trees", "usually nests at some height", or "skulks in undergrowth".

### 3. Habitat variables

A total of five habitat variables (listed in Table 2) were measured at each survey point to characterize the vegetation structure, as this is known to have a role in structuring the avian community [49,50]. These habitat variables were measured in February 2020 halfway through the project period.

### 4. Analysis

**1. Impact on species richness and abundance.** Model selection (AICc) was used to identify the most parsimonious general linear models relating: 1) bird species richness and 2) total avian abundance to distance from each infrastructure type. The full models included distance from each infrastructure type as well as all of the measured vegetation variables. To account for spatial autocorrelation, we calculated Moran's I for the residuals of each model. For models that had a significant Moran's I, we calculated the effective sample size based on this level of spatial autocorrelation and adjusted the p-values accordingly [52].

**2. Impact on bird community composition.** We employed a distance-based redundancy analysis (db-RDA) to examine the relationship between the measured vegetation variables, distance to each infrastructure type, and avian community composition at each survey

**Table 1. Distance categories that were used to spatially stratify the points at which bird surveys were conducted. Ten plots were surveyed in proximity (<1 km) to each infrastructure type/combination.**

| Number of infra-structure types | Infrastructure type/ combination (n = 10 from each type/combination) | Description |
|---|---|---|
| Zero | Control plots | Plots > 1 km from all three infrastructure types |
| One | Road only<br>Rail only<br>Powerline only | Plots within 1 km of only a single infrastructure type |
| Two | Road & Rail<br>Road & Powerline<br>Rail & Powerline | Plots within 1 km of this pair of infrastructure types but not the third infrastructure type |
| Three | Road & Rail & Powerline | Plots within 1 km of all three infrastructure types |

Each survey lasted 20 minutes [44]. The same individual conducted all surveys and all birds either seen or heard were recorded. Birds flying over the plot, above canopy level, were disregarded. All surveys were conducted between 0530 - 0800 h. To avoid any observational bias due to difficulty in hearing birds above the noise of passing vehicles, each survey period was recorded (using a Sennheiser MKE 600 shotgun microphone and a TascamDR-40) and later played back to confirm the recorded species. This study was conducted according to University of Miami Institutional Animal Care and Use Committee (IACUC) protocol 18-160, the Department of Wildlife Conservation in Sri Lanka (WL/3/2/56/17), and the Forest Department in Sri Lanka (R&E/RES/NFSRCM/2018/-02). Additional information regarding the ethical, cultural, and scientific considerations specific to inclusivity in global research is included in the Supporting Information (S1 Checklist).

**Table 2. Vegetation variables measured at each survey point.**

| Variable | Method |
|---|---|
| Canopy cover | 5 photographs pointed directly overhead were taken at 1 m height. Images were analyzed for percent canopy cover using ImageJ software [51] and the mean of the five measurements was recorded. |
| Shrub cover | 5 photographs were taken at 1 m height and oriented horizontally with a 1 x 1 m white board in the background starting from ground level. Images were processed using ImageJ software [51] to determine percent of the white board hidden by shrub cover. The mean of the five measurements was recorded. |
| Herb cover | Five 1 x 1 m quadrats were randomly placed within a 10 m radius from each survey point. The number of 20 x 20 cm sub-quadrats with herb cover was counted, and the mean of the five measurements was recorded as a percentage of total cover. |
| Diameter at Breast Height (DBH) | Number of trees with DBH > 10 cm was counted within a 10 x 10 m plot centered on the survey point. |
| Plant species richness | The number of tree species with DBH > 10 cm within the 10 x 10 m plot centered on the survey point was recorded. |

point. Prior to analysis, we removed singleton species from the community matrix. We used Gower's distance to create the similarity/dissimilarity matrix between survey points. We then employed model selection (via the *ordistep* function in the *vegan* package; [53]) to identify which distance to infrastructure and which vegetation variables best predict avian community composition. Using the resulting ordination, we created two summary axes: 1) an axis pointing toward survey points farthest from all three infrastructure types, and 2) an axis pointing toward survey points far from both road and railroad, but close to powerline. We then reflected each species score on both of these axes to determine their position along these important gradients. Species that fell more than halfway from the origin along either side of the first axis or the positive side of the second axis were identified as those most influenced by distance to LI or LI combinations. To further examine species' placement along the first summary axis, we employed a one-way ANOVA comparing three groups: 1) Sri Lankan endemics, 2) species that have naturalized populations outside of their native range, and 3) all remaining species. The goal of this analysis was to assess whether proximity to LI affects first, the occurrence of avifauna that are most unique to this portion of the globe, and second, the occurrence of avifauna that are most shared with other portions of the globe. All statistical analyses were performed in R ver 3.6.0 [54] or JMP Pro 15 [55].

## 3. Results

Across the 80 survey points and three survey periods, we recorded a total of 95 bird species belonging to 42 families, including 11 migratory species (Supplementary Table 1). Among these species, there were eight endemics (i.e., Sri Lanka Grey Hornbill, *Ocyceros gingalensis*; Sri Lanka Woodshrike, *Tephrodornis pondicerianus*; Pompadour Green-pigeon, *Treron pompadora*; Sri Lanka Junglefowl, *Gallus lafayetii*; Layard's Flameback, *Chrysocolaptes stricklandi*; Black-capped Bulbul, *Pycnonotus melanicterus*; Crimson-fronted Barbet, *Megalaima rubricapilla* and Sri Lanka Brown-capped Babbler, *Pellorneum fuscocapillum*). We also found one Vulnerable species (Besra, *Accipiter virgatus*) and seven Near Threatened species (Grey-headed Fish-eagle, *Ichthyophaga ichthyaetus*; Oriental Dwarf Kingfisher, *Ceyx erithacus*; Banded Bay Cuckoo, *Cacomantis sonneratii*; Drongo Cuckoo, *Surniculus lugubris*; Thick-billed Flowerpecker, *Dicaeum agile*; Great Racket-tailed Drongo, *Dicrurus paradiseus*; and Brown Wood Owl, *Strix leptogrammica*) [46]. Eight species had naturalized and invasive populations in other parts of the world. These species include: Spotted Dove (*Streptopelia chinensis*),

Scaly-breasted Munia (*Lonchura punctulata*), White-rumped Munia *(Lonchura striata)*, White-rumped Shama (*Copsychus malabaricus*), Indian Peafowl (*Pavo cristatus*), Rose-ringed Parakeet (*Psittacula krameri*), Red-vented Bulbul (*Pycnonotus cafer*), and Common Myna (*Acridotheres tristis*).

## 1. Species richness and abundance are higher close to railroads

Both bird species richness and total avian abundance were higher closer to railroads (p = 0.001 and p = 0.01, respectively). Mean bird species richness ranged from 43 species within 100 m of the railroad to 32 species > 1 km from the railroad (26% reduction). Further, total avian abundance decreased by 20% between the same distance categories. Vegetation indices and distance to roads and powerlines had negligible impacts on bird species richness and total avian abundance. Thus, our first hypothesis was not supported, as both species richness and abundance were higher proximal to LI (or at least railroad) rather than farther away.

## 2. Linear infrastructure homogenizes avian communities

Avian community composition varied significantly with distance to all three types of LI (road: p = 0.001, railroad: p = 0.001, powerline: p = 0.003). This supported our second hypothesis, that distance to LI would shape bird community composition. In contrast, the five vegetation variables did not have a significant effect on community composition. Of the 95 species recorded, we excluded three from the db-RDA because they were only encountered once during the entire survey period: Banded Bay Cuckoo (*Cacomantis sonneratii*), Brown Wood-owl (*Strix leptogrammica*), and Grey-breasted Prinia (*Prinia hodgsonii*).

The resulting ordination revealed that the road and railroad shape avian communities in similar ways (vectors largely parallel), in contrast to the powerline (vector orthogonal) (Fig. 2). This supported our third hypothesis that impacts from LI would vary between those with dynamic impacts (road and railroad) and those with static impacts (powerline). Species' responses to LI fell into three main categories: 1) negatively affected by all three LIs, 2) positively affected by all three LIs, and 3) negatively affected by road and railroad while responding positively to the powerline (Table 3).

Ten species were identified as negatively responding to all LIs, whereas three species were noted as positively responding. Species that responded negatively to the presence of infrastructure tended to be those dependent on interior forest (7 out of 10), while those that responded positively to infrastructure tended to be generalists (3 out of 3). The ten species that were identified as notably avoiding all three LIs also included five endemics. These ten species are the Tickell's Blue Flycatcher (*Cyornis tickelliae*), Black-naped Monarch (*Hypothymis azurea*), endemic Black-capped Bulbul (*Pycnonotus melanicterus*), White-rumped Shama (*Copsychus malabaricus*), endemic Crimson-fronted Barbet (*Megalaima rubricapillus*), Long Billed Sunbird (*Nectarinia lotenia*), endemic Sri Lanka Brown-capped Babbler (*Pellorneum fuscocapillum*), endemic Sri Lanka Grey Hornbill (*Ocyceros gingalensis*), endemic Sri Lanka Junglefowl (*Gallus lafayetii*), and Small Minivet (*Pericrocotus cinnamomeus*). In contrast, the three generalist species identified as most frequently occurring near all LI types were the Red-vented Bulbul (*Pycnonotus cafer*), White-rumped Munia (*Lonchura striata*), and Rose-ringed Parakeet (*Psittacula krameri*). One species, the White-browed Bulbul (*Pycnonotus luteolus*), was identified as preferring habitats close to the powerline while avoiding plots near the road and railroad.

The one-way ANOVA revealed that endemic and non-endemic species differed significantly in terms of their placement along the interior to edge axis (P = 0.002), with endemics

being more likely to occur in interior habitat while non-endemics were more likely to be found in proximity to LI (Fig. 3). The ANOVA also revealed that the eight species known to be naturalized elsewhere around the globe similarly differed (P < 0.001) from the rest of the species in their occurrence along the first summary axis; the naturalized species prefer habitat proximal to LI. These results expanded upon our second hypothesis by identifying endemic species as those most likely to be sensitive to LI and species with naturalized populations as those most likely to be tolerant to LI-generated disturbance.

## 4. Discussion

Our findings demonstrate substantial variation in bird species' response to LIs. While overall species richness and abundance were highest close to LIs, particularly railroad, not all species exhibited this response. In particular, Sri Lankan endemics were found to prefer interior forest habitat distant from LIs. Conversely, species that are known to have naturalized populations in other parts of the world showed a significant preference for habitat proximal to LIs. In combination, these results indicate that LI selects against the unique elements of Sri Lanka's avian community and for those members that are increasingly found at other locations across

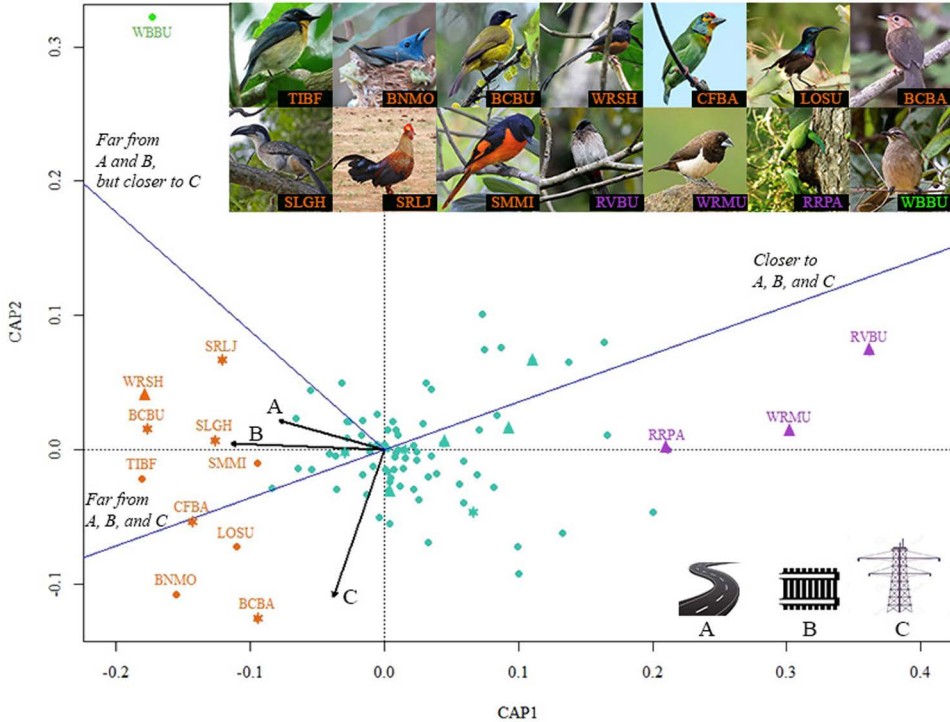

**Fig 2. Ordination plot illustrating bird species' responses to distance from road, railroad, and powerline.** Vectors (black lines) represent distance from each infrastructure type and summary axes (blue lines) show distance to different infrastructure combinations. Species that prefer habitat close to a given LI occur opposite the respective vector and closer to the respective summary axis. (TIBF- Tickell's Blue Flycatcher; BNMO- Black-naped Monarch; BCBU- Black-capped Bulbul; WRSH- White-rumped Shama; CFBA- Crimson-fronted Barbet; LOSU- Long Billed Sunbird; BCBA- Sri Lanka Brown-capped Babbler; SLGH- Sri Lanka Grey Hornbill; SRLJ- Sri Lanka Junglefowl; SMMI- Small Minivet; RVBU- Red-vented Bulbul; WRMU- White-rumped Munia; RRPA- Rose-ringed Parakeet; WBBU- White-browed Bulbul). Orange markers represent species that prefer interior forest, purple markers represent species that prefer proximity to LI, and green markers represent species that specifically favor the powerline but not the other two LI types; endemic species are denoted with asterisks and species with naturalized populations around the globe are denoted with triangles.

**Table 3. Bird species that are notably associated with distance to linear infrastructure. Dependency on interior forest is based on habitat descriptions from Warakagoda et al. (2012).**

| Family | Species | Dependency on interior forest habitat | Road | Railroad | Powerline |
|---|---|---|---|---|---|
| Muscicapidae | Cyornis tickelliae (Tickell's Blue Flycatcher) | High | Far | Far | Far |
| Monarchidae | Hypothymis azurea (Black-naped Monarch) | High | Far | Far | Far |
| Pycnonotidae | Pycnonotus melanicterus (Black-capped Bulbul)* | High | Far | Far | Far |
| Muscicapidae | Copsychus malabaricus (White-rumped Shama) | High | Far | Far | Far |
| Ramphastidae | Megalaima rubricapillus (Crimson-fronted Barbet)* | Low | Far | Far | Far |
| Nectariniidae | Nectarinia lotenia (Long Billed Sunbird) | Low | Far | Far | Far |
| Timaliidae | Pellorneum fuscocapillus (Sri Lanka Brown-capped Babbler)* | High | Far | Far | Far |
| Bucerotidae | Ocyceros gingalensis (Sri Lanka Grey Hornbill)* | High | Far | Far | Far |
| Phasianidae | Gallus lafayetii (Sri Lanka Junglefowl)* | High | Far | Far | Far |
| Campephagidae | Pericrocotus cinnamomeus (Small Minivet) | Low | Far | Far | Far |
| Pycnonotidae | Pycnonotus cafer (Red-vented Bulbul) | Low | Near | Near | Near |
| Estrididae | Lonchura striata (White-rumped Munia) | Low | Near | Near | Near |
| Psittacidae | Psittacula krameri (Rose-ringed Parakeet) | Low | Near | Near | Near |
| Pycnonotidae | Pycnonotus luteolus (White-browed Bulbul) | Low | Far | Far | Near |

*endemic.

the globe, leading to biotic homogenization. We also identified differential effects of roads and railroads (dynamic impacts) vs. powerline (static impact) on avian community composition.

Three previous studies have examined the impact of railroads on bird communities and all three found similar positive impacts on species richness and abundance. All three studies were conducted in Poland, two in a pine forest habitat [56,57] and the other in an agricultural landscape [58] (Kajzer-bonk *et al.*, 2019). It is interesting that the same pattern holds in a tropical dry-mixed evergreen forest as was observed in these temperate habitats. These observations could be attributed to the greater availability of additional resources at the habitat edge with a relatively low risk for accessing them (i.e., lower traffic at railroads than at roads, resulting in a lower collision risk and disturbance; [5,57,59]Borda-de-Água et al., 2017; Rytwinski and Fahrig, 2012; Wiacek et al., 2015). Railroads attract birds because railroad-associated infrastructure act as excellent song, look-out, or resting posts safe from the attention of predators [12,60], and open terrain along the tracks provides good foraging habitats [12,61].

In contrast to the positive effects of railroads, previous studies of road effects on avian communities have found lower species richness and abundance in the vicinity of the road edge compared to interior forest plots [62,63]. While we did not see a negative effect of roads, we also did not observe a positive effect as we did for railroad. While some of the previous studies predict road noise to be the main driver for such negative trends [62,64], others point to road mortality as the main cause [65]. Again, most of these studies have occurred in temperate regions [66] and hence may not be the case for tropical regions. However, a comparable study conducted in a dry evergreen forest landscape in northeastern Thailand reports that both species richness and abundance increased with distance from the edge, but this relationship was only marginally significant for abundance [67]. This variance in species response to LI development emphasizes the importance of not green lighting LI development activities in tropical regions just because data is limited.

We discovered a wide divergence in how the avian community responds to dynamic (roads and railroads) vs. static impacts (powerlines) as illustrated by their largely orthogonal vectors in the ordination. Regular vehicular movement on roads and railroads generate an array of

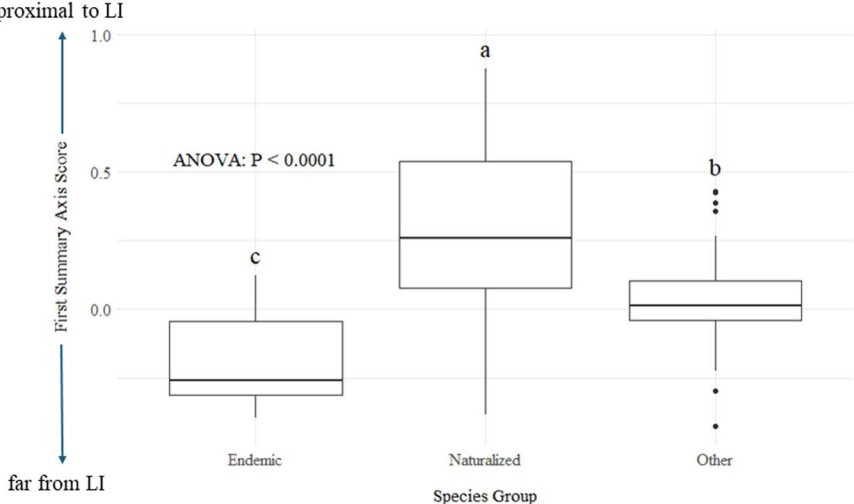

**Fig 3. Box plot comparing species groups (Endemic, Naturalized, and Other) based on their positions along the First Summary Axis, representing the gradient from interior forest (low values) to sites proximal to LI (high values).** Groups with different letters vary significantly in their placement along this axis.

dynamic impacts on the environment including animal-vehicle collisions, disturbance from noise and light, and pollution from vehicle emissions [4,5,68]. Therefore, roads and railroads affect animal populations by causing direct mortality [69] in addition to triggering certain behavioral responses such as avoiding noisy or night-lit habitats [70,71] and altering predation rate at edges [23,72]. Further, the nature of the verge created by a road or railroad is different from that created by a powerline. For roads and railroads, it is necessary to keep the ground of the linear corridor either paved or completely cleared to ensure safe transportation. However, for powerlines it is only necessary to maintain the canopy at a certain height below the cables to ensure uninterrupted power transmission. Therefore, the linear corridor of a powerline is frequently left as a strip of grass or shrub. These differential impacts may explain why the avian community present in the Sri Lankan dry forest responds differently to divergent LIs.

One species with a particularly distinct response to static vs. dynamic LI was the White-browed Bulbul. During the present study, it was observed frequently in plots close to the powerline while avoiding plots near the road and railroad. This observation matches this species' habitat preference (i.e., scrub, forest understory, and shrubby gardens; [48]), which are representative of the dense, short canopy habitat maintained in powerline corridors. Other studies that have investigated the impact of similar transmission corridors on birds report that these habitats have an immense conservation importance for scrub/shrub birds, as these corridors are maintained in an early stage of succession and hence serve as ideal scrub/shrub habitat [73,74]. The example of the White-browed Bulbul emphasizes the importance of considering differential impacts of LI types.

Overall response to LI showed wide variation between bird species. This could be because different guilds respond differently to the edges created by LIs. For instance, in a study conducted on edge effects in remnant rainforest patches embedded in two matrix types (mining vs. agricultural) on birds in southwest Ghana, forest specialists were the most negatively affected group [75]. A study conducted in Thailand's dry-mixed evergreen forest revealed that the avian guilds most negatively affected by road induced edge effects include understory insectivores, arboreal frugivore-insectivores, and raptors [67]. In addition, an experimental study showed that ground nesting birds with low-frequency calls are most prone to negative effects of vehicular movement [63]. Further, a meta-analysis conducted to understand the link between life history traits and population responses to roads asserts that more mobile birds and birds with larger territories are more susceptible to negative road and/or traffic effects than their counterparts [59]. It is thus unsurprising that forest specialists (Table 3), insectivores (Tickell's Blue Flycatcher, Black-naped Monarch, Black-capped Bulbul, White-rumped Shama, Brown-capped Babbler, and Small Minivet), ground-nesting birds (Sri Lanka Junglefowl), birds that are more mobile and with larger territories (Sri Lanka Grey Hornbill), and birds with low-frequency calls (Small Minivet, Black-capped Bulbul, Long Billed Sunbird, and Black-naped Monarch) are those most negatively affected by LI in this Sri Lankan dry-mixed evergreen forest.

Our most alarming finding was that five out of the eight endemics recorded during our surveys were in the group of species most negatively affected by LIs, and that overall, endemics were significantly more likely than non-endemics to occur in habitats distant from LIs. The presence of LIs also seems to create favorable habitats for generalist species that are globally recognized for outcompeting native species with their naturalized populations [76–78]. Some of these species are even expanding their habitat within their native ranges, posing a threat to the endemic fauna of Sri Lanka [79,80]. Collectively, these patterns suggest that further LI development could facilitate the spread of generalists and exert greater pressure on endemics, potentially reducing species diversity at both regional and global scales [81], resulting in biotic homogenization [82]. Several studies report similar trends in biotic homogenization

- replacement of endemic species (often forest specialist species) by other ubiquitous species as a result of human development. For instance, Blair [83] demonstrated greater taxonomic overlap between bird communities in urban areas than in rural ones, a pattern evident in both oak woodlands of northern California and eastern broadleaf forests of Ohio. Similarly, Crooks et al. [84] found that avian assemblages in southern California were progressively more similar to those in northern California and Ohio as sites became more urban. As LI is a major component of urbanization, the pattern we observed in our research parallels those found in these other studies. This indicates a negative overall effect of LI on beta diversity, and therefore the 'infrastructure tsunami' predicted to occur in the near future [85] must be tackled with great caution, with a focus on preserving the unique biodiversity of tropical regions.

Our study highlights the importance of considering infrastructure-specific and species-specific responses to LIs rather than focusing solely on overall species richness or abundance when making LI development decisions. While overall species richness and abundance respond positively to LI (at least concerning railroad), which might be used to greenlight future development projects, those species that are the most unique components of Sri Lanka's biota (i.e., endemics) responded negatively, suggesting caution regarding how future LIs are placed. One solution could be to bundle LIs [86], as this will maximize interior forest habitat distant from all LIs, which our study indicates is favorable for a large set of forest-dependent species. Another option is to place LIs at forest edges during the design stage to avoid fragmentation. These options should be sought after first considering alternative sites, especially to avoid constructing LI through sensitive habitats (e.g., habitats occupied by endemics, or protected areas). An understanding of how different biological communities respond to different types of LIs can not only help guide better decisions about placement of LIs, but it can also inform a reassessment of the total amount of LI development that is compatible with biodiversity conservation goals.

## Supporting information

**Supplementary Table 1: Checklist of bird species recorded across the 80 survey points (BrR- Breeding resident; Pro:Endemic- Proposed endemic; WV- Winter visitor; VU- Vulnerable; NT- Near Threatened; LC- Least Concern; NE- Not Evaluated).**
(DOCX)

## Acknowledgments

We thank IDEA WILD for equipment support. Also, we would like to thank our field assistants (especially Mr. Dammithra Samarasinghe for bird call identification, Mr. Himesh Jayasinghe for plant identification, and Mr. Dayananda Banda), the park warden Mr. P. B.B. Madugalle and the field staff of the Minneriya National Park (especially Mr. Rakitha Bandara and Mr. Pathum Rajapakse) for their support. We thank the communities of Moragaswewa and Rotawewa villages for their help.

## Author contributions

**Conceptualization:** Dishane K Hewavithana, Devaka K Weerakoon, Mayuri R Wijesinghe, Christopher A Searcy.

**Data curation:** Dishane K Hewavithana.

**Formal analysis:** Dishane K Hewavithana, Christopher A Searcy.

**Funding acquisition:** Devaka K Weerakoon.

**Investigation:** Dishane K Hewavithana, Mayuri R Wijesinghe.

**Methodology:** Dishane K Hewavithana, Devaka K Weerakoon, Mayuri R Wijesinghe, Christopher A Searcy.

**Resources:** Dishane K Hewavithana, Devaka K Weerakoon, Mayuri R Wijesinghe.

**Supervision:** Devaka K Weerakoon, Christopher A Searcy.

**Validation:** Devaka K Weerakoon, Christopher A Searcy.

**Visualization:** Dishane K Hewavithana, Christopher A Searcy.

**Writing – original draft:** Dishane K Hewavithana.

**Writing – review & editing:** Devaka K Weerakoon, Mayuri R Wijesinghe, Christopher A Searcy.

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
