## [Decision Letter · Decision Letter 0]

28 Aug 2024

PONE-D-24-14498Linear infrastructure drives biotic homogenization among bird species of a tropical dry forestPLOS ONE

Dear Dr. Searcy,

Thank you for submitting your manuscript to PLOS ONE. After careful consideration, we feel that it has merit but does not fully meet PLOS ONE’s publication criteria as it currently stands. Therefore, we invite you to submit a revised version of the manuscript that addresses the points raised during the review process.

Please submit your revised manuscript by Oct 12 2024 11:59PM,  If you will need more time than this to complete your revisions, please reply to this message or contact the journal office at plosone@plos.org . Please include the following items when submitting your revised manuscript:

We look forward to receiving your revised manuscript.

Kind regards,

Tunira Bhadauria, Ph.D.

Academic Editor

PLOS ONE

**Journal Requirements:**

We thank the National Research Council of Sri Lanka for financial support (Grant No. 16-029) and IDEA WILD for equipment support. Also, we would like to thank our field assistants (especially Mr. Dammithra Samarasinghe for bird call identification, Mr. Himesh Jayasinghe for plant identification, and Mr. Dayananda Banda), the park warden Mr. P. B.B. Madugalle and the field staff of the Minneriya National Park (especially Mr. Rakitha Bandara and Mr. Pathum Rajapakse) for their support. We thank the communities of Moragaswewa and Rotawewa villages for their help.

Dr. Devaka K. Weerakoon received funding from the National Research Council of Sri Lanka. Grant Number 16-029. The funders did not play any role in the study design, data collection and analysis, decision to publish, or preparation of the manuscript.

5. Please note that your Data Availability Statement is currently missing the direct link to access each database. If your manuscript is accepted for publication, you will be asked to provide these details on a very short timeline. We therefore suggest that you provide this information now, though we will not hold up the peer review process if you are unable.

6. PLOS requires an ORCID iD for the corresponding author in Editorial Manager on papers submitted after December 6th, 2016. Please ensure that you have an ORCID iD and that it is validated in Editorial Manager. To do this, go to ‘Update my Information’ (in the upper left-hand corner of the main menu), and click on the Fetch/Validate link next to the ORCID field. This will take you to the ORCID site and allow you to create a new iD or authenticate a pre-existing iD in Editorial Manager.

Reviewers' comments:

Reviewer's Responses to Questions

**Comments to the Author**

1. Is the manuscript technically sound, and do the data support the conclusions?

Reviewer #1: Yes

Reviewer #2: Yes

Reviewer #3: Partly

2. Has the statistical analysis been performed appropriately and rigorously? 

Reviewer #1: N/A

Reviewer #2: Yes

Reviewer #3: No

3. Have the authors made all data underlying the findings in their manuscript fully available?

Reviewer #1: No

Reviewer #2: No

Reviewer #3: Yes

4. Is the manuscript presented in an intelligible fashion and written in standard English?

Reviewer #1: Yes

Reviewer #2: Yes

Reviewer #3: Yes

5. Review Comments to the Author

**Reviewer #1:**  Comments/Suggestions:

How was species richness measured? Please mention in methodology section.

Only species richness /abundance cannot be assumed as a good indicator of measuring impacts of developmental projects/LIs. There is a necessity of searching out and reporting feeding, nesting and breeding sites close to these LIs. Moreover, there is a need to look out for species-specific viable populations close to LIs.

What kind of fuel is used to drive trains? Was there any impact on surrounding air quality due to the fuel?

There are several reports of bird mortality due to collision with powerlines from different parts of the world. Was bird mortality observed and reported near LIs during surveys?

Data was collected in two seasons. Where is the seasonal variation in bird species richness shown in the ordination graph? Under “Bird Surveys” section, the span of dry and wet seasons is incorrectly mentioned.

Mention the summary of the significant results on the axes of ordination plot with the help of a Table.

**Reviewer #2: ** I have read the manuscript " Linear infrastructure drives biotic homogenization among bird species of a tropical dry forest". The authors evaluated the association of the bird community with three types of linear infrastructure within a tropical dry forest. They tested the effect of vegetation structure on bird community richness, abundance and composition.

The document is very well written, presents a very clear, complete and relevant experimental design.

There is only one point that seems to me to require further clarification. That is, the authors indicate that they performed an ANOVA “to determine if there was a difference between endemics and non-endemics in terms of their placement along the first summary axis generated above”, did they use the coordinates of each species along the gradient to perform the analysis, or did they use the abundance values? If the latter, perhaps it would be better to perform a GLM poisson (log) to make the comparison

It was a great pleasure to read it.

**Reviewer #3: ** The manuscript assesses the effects of linear infrastructures, such as roads, railroads and powerlines, on species richness, abundance, and composition of bird assemblages in Sri Lanka. I found the manuscript well written, however, crucial information is missing regarding how data was analyzed. Also, the concept of biotic homogenization is not properly addressed in the manuscript. Please find bellow my specific comments.

Hypotheses could be drawn based on the information provided in the introduction. What are the expectations?

The title of the manuscript indicates a spatio-temporal process of decreasing betadiversity (biotic homogenization); however, such process is not addressed in the introduction. In fact, the potential detrimental effects of LI may vary through time which could be linked in the introduction. Betadiversity is not mentioned at all.

Page 6: “Third, the selected site has all three LIs of interest (i.e., road, railroad, and powerline) in close proximity to each other within the same forested landscape.” Sites could present spatial autocorrelation? Did the authors consider the lack of independence (spatially and temporally) of sampling units in the analyses?

What is the number of plots within each LI type? Table 1 could show the number of plots in each combination of LI.

Is the sampling adequate to represent species abundance? Perhaps the usage of incidence data would be less prone to biases.

The authors present a very short description of models, it is important to add more details such as type of model, assumptions, and combination of variables.

Endemics could be naturally rare; how did the authors consider such potential bias? I am not sure if the sampling is adequate to make inference about rare species, 20 min of observation in each point during three survey periods is potentially recording only common species.

How was endemism determined? Based on which source? Please provide details and reference.

Overall, the discussion is well written, however, I miss a more in-depth discussion of biotic homogenization. Central to this discussion is understanding that this process is fundamentally temporally and spatially explicit. The authors need to plan, analyze and discuss their results with that in mind. Is not only about endemic/non-endemic, is about how species with narrow distributions are replaced by widely distributed species across regions and over time, leading to a reduction of betadiversity. It is also a process that is mostly assessed at large scales.

6. PLOS authors have the option to publish the peer review history of their article (what does this mean? ). If published, this will include your full peer review and any attached files.

**Do you want your identity to be public for this peer review?** For information about this choice, including consent withdrawal, please see our Privacy Policy .

Reviewer #1: **Yes: ** Dr. Bindia Gupta

Reviewer #2: No

Reviewer #3: No

---

## [Author Response · Author response to Decision Letter 1]

27 Oct 2024

Dr. Tunira Bhadauria

October 11, 2024

Dear Dr. Bhadauria,

Thank you for facilitating the review of our manuscript titled " Linear infrastructure drives biotic homogenization among bird species of a tropical dry forest." We truly value your time and effort in assessing our work.

We have addressed all the comments as outlined below (italicized). Additionally, these adjustments have been tracked through Track Changes to ensure easy identification within the manuscript.

Reviewer comments and author responses:

Reviewer #1:

Comment 1: How was species richness measured? Please mention in methodology section.

Response: Thank you for your suggestion. We have now clarified how species richness was measured in the methodology section. Specifically, we have added that species richness was estimated based on the incidence (presence/absence) of species recorded at each plot, as referenced in Totterman (2012).

Comment 2: Only species richness /abundance cannot be assumed as a good indicator of measuring impacts of developmental projects/LIs. There is a necessity of searching out and reporting feeding, nesting and breeding sites close to these LIs. Moreover, there is a need to look out for species-specific viable populations close to LIs.

Response: Thank you for your insightful comment. We agree that assessing feeding, nesting, and breeding sites, as well as monitoring species-specific viable populations near linear infrastructures (LIs), are important indicators for evaluating the impact of development projects. These additional factors could indeed be incorporated as further steps based on the findings of the current study. However, the primary focus of this research is on species richness and abundance, which serve as initial metrics for understanding broader ecological impacts.

Comment 3: What kind of fuel is used to drive trains? Was there any impact on surrounding air quality due to the fuel?

Response: Thank you for your question. In Sri Lanka, trains are powered by diesel fuel. While no significant impact on air quality has been observed, a thorough investigation would be required to confirm the actual status. However, the presence of lichens on tree trunks in the area suggests that the surrounding ecosystem is not significantly affected by train emissions. Additionally, the relatively low frequency of train traffic may contribute to minimizing the impact on air quality.

Comment 4: There are several reports of bird mortality due to collision with powerlines from different parts of the world. Was bird mortality observed and reported near LIs during surveys?

Response: No bird mortality was recorded near the high-tension powerlines during our surveys. The transmission wires are well separated, allowing birds to fly through them without issue. Additionally, the bird species in the area have relatively small wingspans, reducing the risk of collision. While regular powerlines with closely spaced wires can pose a threat to birds, the high-tension powerline studied here, with its ample gaps, did not present such risks.

Comment 5: Data was collected in two seasons. Where is the seasonal variation in bird species richness shown in the ordination graph? Under “Bird Surveys” section, the span of dry and wet seasons is incorrectly mentioned.

Response: Thank you for your thoughtful comment. We appreciate your attention to this detail.

In response, we would like to clarify that the objective of sampling across seasons was to capture a comprehensive picture of bird abundance and species richness, rather than to compare the dry and wet seasons. As for the correct span of the dry and wet seasons, we think we understand the source of the confusion. The full duration of the dry and wet seasons in mentioned in the previous paragraph, whereas in the bird surveys section we are specifically stating which months our own surveys occurred in. We have clarified this intent in the manuscript.

Comment 6: Mention the summary of the significant results on the axes of ordination plot with the help of a Table.

Response: Thank you for your comment. We agree that presenting the significant findings from the ordination plot in a clearer format would benefit the reader. As a result, we have now incorporated a box plot summarizing the key findings from the ordination analysis and reference this plot where the relevant results are discussed.

Reviewer #2:

I have read the manuscript " Linear infrastructure drives biotic homogenization among bird species of a tropical dry forest". The authors evaluated the association of the bird community with three types of linear infrastructure within a tropical dry forest. They tested the effect of vegetation structure on bird community richness, abundance and composition.

The document is very well written, presents a very clear, complete and relevant experimental design.

Comment 1: There is only one point that seems to me to require further clarification. That is, the authors indicate that they performed an ANOVA “to determine if there was a difference between endemics and non-endemics in terms of their placement along the first summary axis generated above”, did they use the coordinates of each species along the gradient to perform the analysis, or did they use the abundance values? If the latter, perhaps it would be better to perform a GLM poisson (log) to make the comparison

Response: Thank you for your question. We appreciate your attention to this aspect. We used species score values from the first summary axis for this analysis.

Reviewer #3:

The manuscript assesses the effects of linear infrastructures, such as roads, railroads and powerlines, on species richness, abundance, and composition of bird assemblages in Sri Lanka. I found the manuscript well written, however, crucial information is missing regarding how data was analyzed. Also, the concept of biotic homogenization is not properly addressed in the manuscript. Please find bellow my specific comments.

Comment 1: Hypotheses could be drawn based on the information provided in the introduction. What are the expectations?

Response: We hypothesized that linear infrastructures (LIs), such as roads, railroads, and powerlines, will have a significant impact on bird communities in the tropical dry forest of Sri Lanka. Specifically, we expected that:

• Bird species richness and abundance will decline as proximity to LIs increases, likely due to a combination of direct mortality (roads and railroads) and changes to the abiotic environment (all three LI types).

• Bird community composition will vary with distance from LIs, with disturbance-tolerant species found closer to the infrastructures and more sensitive species present farther away.

• The type of LI will influence the magnitude of the impact, with roads and railroads having more similar effects compared to powerlines, which are expected to have a lesser impact due to their lack of moving vehicles.

We have now stated these hypotheses clearly at the end of the Introduction and revisited them at appropriate points in the Results to indicate whether or not they received support from the data.

Comment 2: The title of the manuscript indicates a spatio-temporal process of decreasing betadiversity (biotic homogenization); however, such process is not addressed in the introduction. In fact, the potential detrimental effects of LI may vary through time which could be linked in the introduction. Betadiversity is not mentioned at all.

Response: Thank you for pointing out this shortcoming. We have added a description of biotic homogenization to the Introduction, highlighting that it is often driven by habitat fragmentation, which is one of the hallmarks of LI.

Comment 3: Page 6: “Third, the selected site has all three LIs of interest (i.e., road, railroad, and powerline) in close proximity to each other within the same forested landscape.” Sites could present spatial autocorrelation? Did the authors consider the lack of independence (spatially and temporally) of sampling units in the analyses?

Response: Thank you for pointing out this concern. We have checked the residuals from our general linear models and discovered that there is significant autocorrelation in our model for bird species richness, but not in our model for total avian abundance. Based on the Moran’s I value for our species richness model, we calculated our effective sample size and adjusted our p-values accordingly. The effective sample size was reduced 37% compared to the actual sample size, but given that the effective sample size was still relatively large (50), the corresponding p-value still rounded to P = 0.001.

Comment 4: What is the number of plots within each LI type? Table 1 could show the number of plots in each combination of LI.

Response: Thank you for your suggestion. In response, we have clarified that each combination of LI types were represented by 10 plots in the present study. This information has now been incorporated into Table 1 as per your recommendation.

Comment 5: Is the sampling adequate to represent species abundance? Perhaps the usage of incidence data would be less prone to biases.

Response: Thank you for your comment. We believe that using the abundance data is more informative, since it is a meaningful difference if up to 28 times as many individuals of a given species are found at one site versus another. Thus, we have kept the species counts in the analysis included in the manuscript. However, to check whether using presence/absence data would change the results, we re-ran the analysis in this manner. The results were almost identical, with no qualitative change and the only quantitative change being that the P-value for powerline was 0.005 instead of 0.003.

Comment 6: The authors present a very short description of models, it is important to add more details such as type of model, assumptions, and combination of variables.

Response: Thank you for your comment. We have added more details about the types of models used, the variables that were being analyzed, and the goal of each analysis. Hopefully these edits make the analyses easier to follow.

Comment 7: Endemics could be naturally rare; how did the authors consider such potential bias? I am not sure if the sampling is adequate to make inference about rare species, 20 min of observation in each point during three survey periods is potentially recording only common species.

Response: Thank you for your insightful comment. It is true that the intensity of sampling we were able to conduct will not detect all species present at a given site. Thus rare species (possibly endemics) are almost certainly present at more sites than just those where we detected them. However, the important thing to note is that we expended equal survey effort at all sampling sites, and thus the species lists generated for each site, while not exhaustive, are unbiased. Moreover, our point count sampling methodology, which involved 20-minute observations at each site repeated over three survey periods, aligns with methods used in previous studies conducted in the park (Bibby and Burgess, 1993; DWC, 2008), which is why we had an expectation that this survey method would be sufficient for us to detect the patterns we were striving to investigate. As it turned out, our expectation was correct, as this intensity of survey effort was sufficient for us to detect significant patterns in avian abundance, species richness, and community composition.

Bibby, C.J., Burgess, N.D., 1993. Bird census techniques, Trends in Ecology & Evolution. Academic Press Limited, London. https://doi.org/10.1016/0169-5347(93)90208-7

DWC, 2008. Biodiversity Baseline Survey: Minneriya National Park. Consultancy Services Report. Infortech IDEAS in association with GREENTECH Consultants.

Comment 8: How was endemism determined? Based on which source? Please provide details and references.

Response: Thank you for your question. Endemism was determined using the National Red List of Sri Lanka, which provides an extensive list of endemic and native fauna and flora species based on a comprehensive analysis of field data collected by experts. This list is prepared according to IUCN guidelines and protocols. In this study, we used the bird species list from the National Red List of Sri Lanka as our source to determine species endemism. The bird species names were adopted from the publication by the Ministry of Environment (MoE), 2012: The National Red List of Sri Lanka: Conservation Status of the Fauna and Flora, Colombo.

We had previously referred to this source in the results section; however, for clarity, we have now added a clarifying line and cited the same source in the Bird Surveys section.

Comment 9: Overall, the discussion is well written, however, I miss a more in-depth discussion of biotic homogenization. Central to this discussion is understanding that this process is fundamentally temporally and spatially explicit. The authors need to plan, analyze and discuss their results with that in mind. Is not only about endemic/non-endemic, is about how species with narrow distributions are replaced by widely distributed species across regions and over time, leading to a reduction of betadiversity. It is also a process that is mostly assessed at large scales.

Response: Thank you for your valuable comment. We acknowledge that our data is collected during a single time period, and thus we are inferring a temporal process based on the spatial pattern that we observe during this snapshot. In particular, we observe wide ranging species (those that have been naturalized across the globe) appearing close to linear infrastructure, while species with narrow distributions (endemic species) are observed at sites distant from linear infrastructure. Since the linear infrastructure is a relatively new addition to this dry forest, we thus infer a temporal process in which avian communities that were formerly more unique are being replaced by avian communities that increasingly resemble those found at other locations across the globe, thus decreasing beta diversity not within the tropical dry forest, but between the tropical dry forest and those other sites around the globe (i.e., at a large geographic scale). We have tried to clarify this line of reasoning during our revision. We have also added an additional analysis so that we are not just analysing endemics vs. non-endemics, but are also comparing between species that have increasingly wide distributions around the globe (naturalized species) and those that do not.

End of editor and reviewer comments---------------------------------------------

Journal Requirements:

Amended funding statement: Dr. Devaka K. Weerakoon and Dishane Kalya Hewavithana received funding support from the National Research Council of Sri Lanka (Grant Number 16-029). The funders had no role in study design, data collection and analysis, decision to publish, or manuscript preparation.

Amended data availability statement: We would like to confirm that we will be making the data available through Dryad upon acceptance of the manuscript. We have selected Dryad as the repository, which complies with your open data policy and ensures that the data will be freely accessible. We will ensure that the data deposition process is completed promptly to avoid any delays in the publication timeline.

An ORCID iD for the corresponding author is added in Editorial Manager: We have made this change in Editorial Manager as requested.

Please include your full ethics statement in the ‘Methods’ section of your manuscript file. In your statement, please include the full name of the IRB or ethics committee who approved or waived your study, as well as whether or not you obtained informed written or verbal consent. If consent was waived for your study, please include this information in your statement as well.

A statement of the Institutional Animal Care and Use Committee approval for this study is included in the Bird Surveys section of the Methods. Permits from the Sri Lankan Department of Wildlife Conservation and Forest Department are listed as well.

Please include captions for your Supporting Information files at the end of your manusc

---

## [Editor Report · Decision Letter 1]

11 Feb 2025

Linear infrastructure drives biotic homogenization among bird species of a tropical dry forest

PONE-D-24-14498R1

Dear Dr. Searcy

We’re pleased to inform you that your manuscript has been judged scientifically suitable for publication and will be formally accepted for publication once it meets all outstanding technical requirements.

Kind regards,

Tunira Bhadauria, Ph.D.

Academic Editor

PLOS ONE
---

## [Editor Report · Acceptance letter]

PONE-D-24-14498R1

PLOS ONE

Dear Dr. Searcy,

I'm pleased to inform you that your manuscript has been deemed suitable for publication in PLOS ONE. Congratulations! Your manuscript is now being handed over to our production team.

Kind regards,

on behalf of

Dr. Tunira Bhadauria

Academic Editor

PLOS ONE